# Recent Advances in Applications of Carbon Nanotubes for Desalination: A Review

**DOI:** 10.3390/nano10061203

**Published:** 2020-06-19

**Authors:** Ruiqian Wang, Dinghao Chen, Qi Wang, Yibin Ying, Weilu Gao, Lijuan Xie

**Affiliations:** 1College of Biosystems Engineering and Food Science, Zhejiang University, Hangzhou 310058, China; rqwang@zju.edu.cn (R.W.); chendinghao@zju.edu.cn (D.C.); wangq13015@zju.edu.cn (Q.W.); ibeying@zju.edu.cn (Y.Y.); 2Key Laboratory of on Site Processing Equipment for Agricultural Products, Ministry of Agriculture and Rural Affairs, Zhejiang University, Hangzhou 310058, China; 3Zhejiang A&F University, Hangzhou 311300, China; 4Department of Electrical and Computer Engineering, Rice University, Houston, TX 77251-1892, USA; wg6@rice.edu

**Keywords:** carbon nanotube (CNT), membrane, desalination, mechanism, effect

## Abstract

As a sustainable, cost-effective and energy-efficient method, membranes are becoming a progressively vital technique to solve the problem of the scarcity of freshwater resources. With these critical advantages, carbon nanotubes (CNTs) have great potential for membrane desalination given their high aspect ratio, large surface area, high mechanical strength and chemical robustness. In recent years, the CNT membrane field has progressed enormously with applications in water desalination. The latest theoretical and experimental developments on the desalination of CNT membranes, including vertically aligned CNT (VACNT) membranes, composited CNT membranes, and their applications are timely and comprehensively reviewed in this manuscript. The mechanisms and effects of CNT membranes used in water desalination where they offer the advantages are also examined. Finally, a summary and outlook are further put forward on the scientific opportunities and major technological challenges in this field.

## 1. Introduction

Efficient and sustainable desalination technologies have developed under the pressure of water scarcity [1,2]. In particular, the interest in membrane-based methods among researchers and policymakers is increasing due to their superior and distinct advantages, such as high water quality with easy maintenance [3], compact modular construction [4], low chemical sludge effluent [5] and excellent separation efficiency [6]. According to the membrane process, membranes can be generally divided into different categories such as microfiltration (MF), ultrafiltration (UF), nanofiltration (NF), reverse osmosis (RO), forward osmosis (FO), pervaporation and membrane distillation [7]. Inorganic and polymeric material membranes have drawn much research attention in the past years. However, most of them are restricted by some demerits such as high fouling tendency and a tradeoff between permeability and selectivity [8,9,10,11]. These problems will reduce permeability and membranes’ lifespan due to increasing transmembrane pressure and then become the challenges of desalination [11]. Recently, with the development of nanotechnology, novel materials with well-defined nanostructures [12], such as zeolites, metal–organic frameworks and carbon-based materials, are surfacing in desalination owing to their excellent separation performance and reliable processing strategies [13]. Among them, carbon-based nanomaterials, especially carbon nanotubes (CNTs) [14,15] are promising due to the large surface area, ease of functionalization, improved antifouling behaviors, superb mechanical strength, excellent sieving capabilities and exceptional water transport properties [7,9,16,17]. In general, CNTs can be applied either as direct filters or as a filler to enhance desalination performance [7]. Experimental evidence of mass transport in CNT was first reported in 2004 [18]. While the development of CNT membranes is progressing at a steady pace in recent years, various computational and experimental studies have been conducted to study CNT membranes to remove ions with different characteristics [19]. As the study reported, the water flux through the inner wall of the CNTs was estimated to be 3–4 orders of magnitude higher than predicted by the Hagen–Poiseuille equation [20]. Despite years of research into CNT channels, most of the studies for CNT nanochannels are based on simulations, because there are some obstacles to preparing large-scale vertically aligned CNT membranes [21]. The recent development of nanomaterials and nanotechnology have opened a way for the development of scaling up CNT membranes [19]. For composited CNT membranes, the structure and function of the membrane will also be changed after adding CNTs to the membrane, improving the desalination performance of membranes [22]. For example, CNTs provide membrane stability and resistance to chlorine because of the ability to prevent the polyamide (PA) from reacting with the chlorine [23]. The addition of CNTs will offer many new physical/chemical properties to the composited CNT membranes [7].

From the aforementioned studies, CNTs possess enormous potentials that can advance the membranes with improved capabilities for desalination. There are numerous works [6] dedicated to the preparation of CNT membranes for desalination. Some reviews with different emphasis have summarized the existing research [7]. However, there needs to be a comprehensive review which carbon nanotubes are used for desalination and how to improve their performance. Therefore, this study reviews the applications of CNTs based on MF, UF, NF, RO and FO in desalination from the perspective of mechanism. Moreover, it will provide helpful information to researchers in a bid to advance the CNT membrane development of desalination applications. In this review, first, the mechanisms and influence factors of CNTs used for water desalination are thoroughly introduced, which are systematically analyzed by examining three aspects: the characteristics of CNTs affecting the water flow rate, the external conditions and others affecting the water flow rate of CNTs. Influence factors of CNTs affecting water flux such as diameter, length, modification, etc. are mainly demonstrated. Then, the effects on membranes exerted by adding CNTs are summarized in four parts, including water permeability and flux, ion selectivity, fouling resistance and mechanical performance. Finally, due to the challenges of cost, scale-up, stability, reproducibility, unclear mechanism and environmental impact, we outline the significant opportunities in this field that need to address real-world applications.

## 2. CNT Membranes for Reverse Osmosis Desalination

CNTs, their extremely long graphitic channels with nanoscale diameter have gained compelling attention for the preparation of inventive membranes for water desalination [7]. In general, CNT membranes can be divided into two categories: freestanding CNT membranes and composited CNT membranes [7,24]. Their representative images are displayed in Figure 1 [24]. The two main types of freestanding CNT membranes commonly used for desalination are vertically aligned CNT (VACNT) membranes (Figure 1a) and buckypaper CNT membranes (Figure 1b) [25,26]. Composited CNT membranes mainly include three categories: mixed-matrix CNT membranes (Figure 1c), CNTs as the intermediate layer (Figure 1d), CNTs coated on the membrane surface or support (Figure 1e) [24]. The structure of these composited CNT membranes is similar to that of thin-film composite (TFC) reverse osmosis (RO) membranes [7]. Moreover, the incorporation of CNTs in membranes is capable of achieving an improved fouling resistance, desalination performance, water flux and other purposes [27]. There are other advantages of composite CNT membranes, including simple synthesis procedures and large-scale applications in comparison with VACNT membranes [7]. In this review, we will focus on the introduction of the mechanisms and the influence factors of CNTs to enhance the ability for desalination.

### 2.1. Mechanisms and Influence Factors for Desalination

For CNTs, owing to the violation of the no-slip boundary conditions, the water penetration through CNTs is several orders of magnitude faster than the upper limit predicted by the Hagen–Poiseuille equation [28,29]. Chan et al. may reveal the mystery of the unexpected high flow rate in CNTs and attribute it to the suction generated at the tube entry [30]. In a research conducted by Ma et al., they found the coupling between confined water molecules and the longitudinal phonon modes of CNTs could enhance the confined water diffusion by more than 300% [31]. The increase in diffusion coefficient of water inside CNTs can also be explained by the almost negligible interaction of water molecules with CNT walls [32]. Although different methods implemented to control temperature causes the disparity in simulation data, these results support the claims of extreme and near-frictionless water flow in CNTs [33,34]. To improve the ability of CNT membranes in desalination, it is essential to explore the influence factors mainly derived from the characteristics of the CNTs and the external conditions (Figure 2) and analyze how these factors affect the flow of water molecules in CNTs. Typically, these characteristics include the diameter, length, shape, italicized angle, modification, and the state of the pores in the tube wall. The conditions include electric field, pressure, centrifugal force, etc.

#### 2.1.1. The Characteristics of CNTs Affecting the Water Flow Rate

• The diameter of CNTs

For VACNT membranes, the water transport is highly diameter-dependent inside CNTs [17,39,40,41]. Considering the high flow barriers at the inlet and outlet regions of small diameter CNTs, the flow rate and the flux of water molecule across the CNTs will decrease sharply with the decrease of CNT’s diameters [42]. For CNTs with small diameter, the size of the fluid particles becomes crucial and takes up a considerable portion in determining the diffusion mobility. But for those with a larger diameter, these effects will diminish [43].

Some experimental data significantly suggest that water configuration in CNTs may vary according to their diameters [44,45]. In medium diameter (1.1–1.2 nm) CNTs, the water turns into an ice-like structure while a bulk-like liquid phase form in large-diameter (≥1.4 nm) CNTs [44]. The behavior of water molecules confined inside single-walled carbon nanotubes (SWCNTs) is entirely different from their bulk analogs. Photoluminescence (PL) spectra support the fact that the quasi-phase transition of the orientational order of the water dipoles occurs in the single-file chain in SWCNTs [46]. Moreover, the study of water molecules escaping into the interior of the large unilamellar vesicles through CNTs proves that sub-nanometer confinement forces water molecules into a single-chain configuration [34]. Even though the increase of CNT diameter brings an increase in the flow [47], some scholars have experimentally demonstrated that the permeability enhancement factor would decrease with the rise of the diameter of carbon nanotubes [48]. Additionally, the flow rate across CNTs is found to be inversely proportional to the water viscosity at different temperatures for CNTs with a diameter greater than 1 nm [42]. The simulation experiment shows that the ideal diameter of the CNTs for desalination is about 1.1 nm with high permeability and ion selectivity [49]. However, it is worth mentioning that the water flow suppression occurs at room temperature, revealing the change in the structure of the confined water governed thermodynamically [50]. It further indicates that the temperature affects the flow rate of water inside CNTs [51,52,53,54]. Liu et al. demonstrated that decreasing the temperature within the (9, 9) CNT can strongly inhibit ion conduction and still permit significant water transport [50]. Nevertheless, the water permeability of the prepared membrane with CNT inner diameter of 3.9 nm increases with temperatures >26 °C [55]. Further research should focus on designing the density of water rationally in the simulation experiment since the water at high-density regimes experiences a structural transition resulting in a dramatical increase in viscosity [56].

• The length of CNTs

There is an optimal channel length for water transport for VACNT membranes [37,57]. CNT membranes will become more permeable if the CNT length is shortened [50]. Moreover, the variation in the length of CNTs confirms the power-law relation of the flow saturation decay [47], which elucidated that the longer the CNTs, the faster the flow rate decays [31].

• The modification of CNTs

CNTs are cylindrically shaped hollowed structure made of a sheet of carbon atoms [58]. CNT is an intrinsically superhydrophobic property, which is beneficial to desalination [59]. Water molecules can transport ultrafast through the highly hydrophobic and smooth inner wall of CNTs [7]. In addition, CNTs have high adsorption capacity due to the high specific surface area and manipulable surface chemistry [59]. However, sodium ions easily bind at the entrance for narrow CNTs [60] as well as are trapped in the interior for larger-diameter CNTs in VACNT membranes, resulting in the blocking of water flow through CNTs [61,62]. Two methods are proposed to prevent the blocking of CNTs by sodium ions while allowing ultrafast water flow with ion rejection: (i) functionalization of the ends of CNTs with functional groups to prevent any direct contact between sodium ions and π-electron-rich aromatic rings of CNTs [63,64,65,66], (ii) application of an electric field from pulling the cations away from the inlets of CNTs [67]. The modification of CNTs not only prevents the blocking of the ions, but also improves the ion rejection performance of the CNTs [68,69,70]. CNTs can be modified in the interior or at the entrance. Compared to the inlet modified CNTs, water flux of the internal modified CNTs is decreasing slightly, but desalination is substantially improved. For example, CNTs encapsulating fullerene (F@CNT) can be considered as a high-performance desalination candidate (Figure 3) [38]. While charged groups added in the interior will block the same charged ions outside but attract oppositely charged ions into the pore. Ideal modification of groups can achieve 100% desalination and obtain high water conductance [71]. To further increase CNTs’ water flux, they should be modified at the same time as higher voltage (at 2 V) is applied [72]. Larger-diameter CNTs (diameter lower than 1.2 nm) can be selected owing to the more effective area for water molecule penetration at the entrance [73]. In this case, researchers should focus more efforts on the preparation of membranes containing functionalized CNTs, meaning that better rejection can be realized under realistic operating pressures [49].

The robust mechanical strength of CNTs makes them ideal as an inorganic disperse phase in mixed-matrix membranes (MMMs), which can reinforce the composite membranes [74]. However, the challenge for mixed-matrix CNT membrane is poor wettability of CNTs [59]. It is difficult for CNTs to incorporate into the polymer matrix uniformly [75]. This difficulty lies mainly in that CNTs are inherently hydrophobic, and the aggregated bundles from van der Waals forces between the CNTs also prevent the CNTs from dispersing in the matrix [59]. Therefore, it is crucial to enhance the dispersibility of CNTs in mixed-matrix CNT membranes [59]. To achieve uniform distribution of CNTs within the polymer matrix, surface modifications or the functionalization of CNTs is a useful technique, which can enhance interfacial interactions to improve the miscibility of CNTs [76,77]. Chemical oxidation by acid treatment such as boiling nitric acid is the most commonly used method for attaching hydroxyl/carboxyl groups to the walls of CNTs [78]. In addition, the dispersibility of CNTs in various polymer matrices can also be enhanced by ozone oxidation [77]. Furthermore, the properties of CNT composited membranes will be improved.

• The Alignment and Shape of CNTs

The italicized alignment of the CNT for one CNT (10, 0) facilitates their applications with better performance of water desalination compared to the vertical alignment [79]. When the italicized angle α was tuned to 73° (α = 0°, vertical), water flux through italicized CNT membranes achieves maximum performance [79]. The main reasons are as follows. First, the area of the high energy status inside CNTs reduces, which enables the movements of water molecules towards CNTs. Second, the free energy difference of water molecules at the entrance of CNTs decreases, favoring water to enter CNTs [79].

Adjusting the carbon nanotube shape could enhance the water permeation rate as well [80]. Razmkhah et al. have compared the effects of different CNTs’ shapes (shown in Figure 3) on water filtration including hetero-junction (Figure 3c), cone-shaped (Figure 3d) and tubular (Figure 3e) CNTs. Cone-shaped CNTs conduct water faster than tubular ones. The higher capacity and lower energy barrier against water passing the cone-shaped CNTs result in shorter residence time and a higher number of water molecules inside CNTs [35]. Moreover, when the water flows from the base to the tip, the channel with 19.2 taper angle (left shown Figure 4a) has the maximum water flux. Inverting the liquid transport direction, the channel with a taper angle of 38.9 (right shown Figure 4a) shows the better water permeability [37]. For real separation applications of cone-shaped CNTs, the separation efficiency is higher when the mixture solution is transported from the tip to the bottom compared with from the bottom to the tip. Because if the opposite transportation happens, the impermeable substance in the solution will block the channel and reduce the efficiency of separation [37].

• The State of Pores in CNT Walls

Based on the molecular dynamics (MD) simulations, super square (6, 6) (SS@(6, 6)) SWCNT networks can act as a new kind of nanoporous membranes. The permeability of SS@(6, 6) SWCNT membranes is much higher than that of nanoporous graphene membranes [82]. Furthermore, the separation of different ions could be controlled by changing the pore size on CNT walls [83]. It is found that the water permeability with 100% salt rejection is both influenced by salt concentration and the pressure on the CNT membrane with different pore sizes. However, the bending and buckling, which lead to enlargement of the pore size of SS@(6, 6) SWCNT under high pressure, are the main reasons for salt ions passing through the tubes. Thus, by optimizing the structural design and adjusting pore size, the strength of the CNT network can be improved, thus obtaining better filtration capability [82].

#### 2.1.2. The External Conditions Affecting the Water Flow Rate of CNTs

Apart from CNTs’ characteristics, water desalination is affected by external conditions [84]. For instance, a high-performance double-walled carbon nanotube (DWCNT) nanochannel for water desalination through electric resonance (applying a strong electrical field 10^6^ V·m^−1^) is achieved [85]. The transportation of ions could be blocked entirely at a specific frequency and amplitude in the MD simulation. Because the alternating electric field induced by the vibrational charge can disrupt the hydration shell of ions inside CNTs through electrical resonance and increase their free energy in water, the ion flow decreases resulting from a high energy barrier inside CNTs. Furthermore, this mechanism is independent of the type of ions and CNTs’ diameters. Therefore, to remarkably improve desalination efficiency larger-diameter CNTs (i.e., CNT with the index (30, 0)) can be used as a VACNT desalination membrane [86]. The application of a hydrostatic pressure gradient for CNTs is also expected as an effective method to achieve large-scale desalination [43]. Additionally, external force to enforce the rotation [21] of CNTs decorated with partial charges can produce water flux through the CNT nanochannels. The MD simulation results imply that when the angular velocity of rotation falls within a specific range (Figure 4b), increasing the angular velocity of CNTs leading to an increase in water flux [36]. For chiral CNTs with pores in the wall, rotating motion (angular velocity ω = 174.5 rad/ns) generates a negative pressure that pulls the brine to the top entrance of the CNT. Once the brine enters the nanochannel, the centrifugal force produced by the rotation throws water molecules out of the pores in the wall. If the pore size is discreetly selected, the ions will all be kept in the CNT. The residual brine in the CNT will naturally discharge from the bottom of the CNT due to the pressure gradient made by the chiral CNT along the rotating axial direction [83].

#### 2.1.3. The Others Affecting the Water Flow Rate of CNTs

Hollow CNT structures usually perform as water permeation channels for desalination. In addition, a slit separated by horizontally stacked CNTs with a neighboring one can also be considered as flow channels, which resembles vertically aligned CNT arrays (Figure 4c) [81]. It indicates that intertube distance is 1 nm of transverse flow CNT membranes for effective desalination regardless of the CNT diameter [81]. MD simulations illustrate that the membrane has excellent desalination performance, and its permeability is more than twice that of atomic-scale graphene slit membranes [87]. Moreover, Ang et al. found that oscillating pressure operation can increase the membrane permeability by 16% and the salt rejection close to 100% [88].

### 2.2. Effects of CNT Addition for Desalination

The properties of composited CNT membranes for desalination will be improved with the addition of CNTs. CNTs have some attractive properties, including high water permeability, improved salt rejection capability, antimicrobial and antifouling properties and mechanical stability [7]. According to the different effects resulting from CNTs, they are illustrated in detail in this section. In addition, we summarized the developed membranes from four aspects in Table 1, including membrane fabrication, the chemical composition of composite membranes, desalination performance and the effects resulting from adding CNTs.

#### 2.2.1. Water Permeability and Flux

After adding CNTs, water permeability is significantly enhanced for VACNT membranes because the water–CNT interaction is negligible. VACNT membranes can be prepared by detaching multiple VACNT films onto substrates [121]. To investigate the effect of different VACNT densities on desalination performance, VACNT membranes were fabricated through spreading liquid PDMS over VACNTs and slicing them into slices [122]. As the density of CNTs increases, the permeability of the membrane increases without sacrificing salt rejection [122,123]. Recently, the study showed that the highest pore density could be achieved by combining volatile ethanol addition and subsequent mold pressing [124]. For VACNT membranes, it is still a challenge to prepare high pore density CNT membranes with a small diameter [125].

CNTs with super-hydrophilicity can be directly blended in the polymer matrix. However, selectivity and mechanical stability will decrease due to the reduced crosslinking of the polymer [100]. In addition, random distribution and a small number of CNTs could not function well for desalination [126]. Furthermore, of note is that the presence of CNTs in the optimum membrane enhances the pure water flux [127,128] with accepted water content percent and swelling percent [91]. For example, the PA layer in combination with the top surface of the high flux outer-wall CNT membrane exhibits an outstanding performance for desalination due to the ultra-dense porosity and hydrophobicity of the outer-wall CNTs [128]. CNT membranes prepared via a phase inversion method [8] also increases the water flux owing to the existence of CNTs [129,130,131]. In a study, by mixing CNT with graphene oxide (GO), CNTs significantly promoted the water flux via adjusting GO interlayer spacing [130]. Furthermore, the CNT hybrid fillers could experience high water permeability using the interfacial polymerization method [97,115,132,133]. Functionalized CNTs can not only easily incorporated into a polymer matrix, but also increase hydrophilicity via hydrogen bonding [132]. Meanwhile, because of its increased hydrophilicity, functionalized CNTs [134] can also enhance water permeability [113,132,135,136,137,138] without sacrificing NaCl rejection [22]. It also shows that the changes in morphology and surface chemistry contribute to the enhancement of permeability (Figure 5a) [105]. To reduce defects and improve stability of CNT membranes, it is necessary to enhance the compatibility between CNTs and polymers [100]. It is reported that the shortened zwitterionization of multiwalled (MW) CNTs (ZCNTs) could probably create a considerable quantity of inorganic–organic interfaces in the barrier layers [120]. ZCNTs also bring about lower energy consumption and enhanced water productivity both for RO [99] and forward osmosis (FO) [120,139]. The preparation of ZCNTs can be initially implemented by grafting copolymers via an atom transfer radical polymerization (ATRP) reaction [120]. To explain for such an improved performance, the mechanism of water diffusion on the nanocomposite CNT membranes was studied, and it was found that the CNTs create a hydrophilic region within the membrane, which may direct water transport across the membrane by generating a low-energy path [102,140]. Moreover, a research compares the structure, property, and performance of the membranes with different CNT locations and the compatibility between CNTs and polymers, indicating that CNTs can offer more channels for water transport [141]. Moreover, the existence of the CNT sublayer is conducive for providing a three-dimensional free space, which can increase the effective area of the active layer [116,142]. To achieve better performance of the CNT membranes, it should be noted that molecular dynamics (MD) is an effective method to direct the preparation of membranes including the reasonable dosage and distribution of CNTs [143].

#### 2.2.2. Ion Selectivity

Generally, the ideal membrane should be thin, durable and able to operate at high salt concentration [100]. The addition of CNTs can also improve ion rejection performance. Moreover, the salt rejection will enhance as the increasing weight percentage of the CNTs in the membrane [100]. It is expected that the pore size of the membrane will decrease and exclusion of Na^+^ and Cl^−^ ions will increase with the addition of CNTs [144]. Further, in addition to increasing water permeation, ZCNTs also improve salt rejection [100]. The steric hindrance instead of charge repulsion to transport caused by the zwitterionic functional groups prevents most ions from penetrating CNTs, leading to high salt rejection [100]. The fabrication process of ZCNTs is shown in Figure 5b. Interestingly, CNTs affects the interfacial polymerization, and the adhesion and dynamics between the CNTs and the matrix are strong, contributing to smaller pore sizes and higher ion rejection of polymer networks [108]. From another point of view, the favorable chemical bonding between CNTs and the PA matrix can also be a cause of enhanced desalination performance [101]. It has been reported that hydrophilic CNTs can disperse well in the aqueous phase and favor of a thin nanocomposite membrane formation of through interfacial polymerization [101]. Tannic acid−Fe^III^-functionalized MWCNTs (TA–MWNTs) with hydrophilicity can be prepared by a green, simple and robust approach presenting a good dispersity [101]. In addition, the results of a study show that the salt rejection of the membrane is improved to 99.8% for the highest content of surface engineered (SE) pristine multi-walled carbon nanotubes (MWCNTs) due to the strong SE–MWCNTs/polymer matrix interaction [112]. However, constrained by permeance–selectivity tradeoff, the MMMs embedded with thinner CNTs have better filtration properties than the thicker ones [93]. Therefore, attention should be paid on the thin CNT membrane with a high water-flux while maintaining salt rejection. Furthermore, improving the dispersion and hydrophilicity of CNTs, and the compatibility between CNTs and polymers is also vital to enhance the desalination properties of CNT membranes.

#### 2.2.3. Fouling Resistance

Furthermore, composited CNT membranes display enhanced fouling resistance [9,94,145,146,147]. Concerning antimicrobial properties, biofouling will decrease [148] owing to the cytotoxic effects of CNTs [149,150]. The overall mechanism of CNT antibacterial activity is caused by their large surface areas and damaging capabilities of the cell membrane by oxidation [151]. In addition, bacteria exposed to CNTs could also cause stress that possibly induces the expression of toxic proteins, which may drastically mutate cellular metabolisms [2,152]. For instance, ZCNTs incorporated into membrane can improve surface biofouling resistance [100]. The ZCNTs appear to be exposed on the membrane surface and interact with the feed water to form a strong hydration layer, thereby improving the surface biofouling resistance [100]. CNTs can also be used in combination with other antibacterial materials to enhance their antibacterial properties [149]. A novel approach is implemented to synthesize a silver-doped CNT membrane [153]. Silver particles serve as a binding material for the CNTs and the produced membrane exhibits strong antibacterial properties [154]. The possible antimicrobial mechanism is that the silver doped CNTs adsorb the bacteria, and removal by the antimicrobial properties of both silver and CNTs [154]. Furthermore, CNTs incorporated to carbon quantum dots exhibited antimicrobial property in the presence of light [155]. As such, to fully utilize the antitoxic properties of nanomaterials, CNTs are much more resistant to biofouling when combined with other antibacterial nanomaterials.

Aside from anti-biofouling properties of CNTs, many studies have been carried out for anti-organic fouling of CNT membranes [98]. The pristine or oxidized MWCNT membranes at different concentrations show better antifouling properties compare to unloaded CNT membranes in 24 h test with bovine serum albumin (BSA) solution [98]. The primary cause is that carboxylated MWCNTs that are dispersedly deposited on the surface of the membrane are used to reduce BSA fouling by forming polymer brushes and hydrodynamic resistance [98]. This kind of antifouling CNT membranes can be simply prepared by grafting of a hydrophilic acrylic acid monomer and carboxylated MWCNTs on commercial RO membrane [98,113]. However, noncovalent interactions between carboxylic CNTs and polymeric matric is weak, which easily leads to poor salt separation effect [114]. Employing ATRP to modify MWCNTs by grating polyacrylamide (PAAm) which can interact strongly with the matric—could be the right choice [114]. Parallel to this, functionalization of MWCNTs with PAAm structure by ATRP has a significant effect on antifouling capability [114]. Antifouling property primarily comes from increasing smoothness of the membrane surface [114]. Due to the negatively charged membrane surface, TFC MMM with functionalized CNTs blended in polyether sulfone (PES) support layer also play essential roles in the interaction enhancement of the repulsive alginate foulant [110]. Specifically, the electrostatic force plays a leading role in membrane organic fouling reduction [110]. Moreover, a novel membrane that CNTs are incorporated into double-skinned thin nanocomposite films demonstrates a remarkable humic acid resistance [104]. The primary mechanism in support of higher organic fouling resistance of these CNT membranes stems from the effect to the change in membrane morphology and changes on the net charge of the membrane surface when CNTs are incorporated into the membrane [22]. Therefore, the preparation of the antifouling CNT membranes is significantly dependent on the smooth membrane surface and the net charge. At the same time, attention should be paid to the CNTs’ dispersibility and interactions with polymeric matrices. Additionally, due to the lower fouling and easier fouling removal of FO membrane, it is also an antifouling desalination membrane development trend.

From another perspective, the addition of MWCNTs decreases ion adsorption [156], providing the membrane with outstanding chlorine resistance [23,141,157]. CNT incorporated to PA membrane exhibits lower absorption of chlorine [23]. However, the mechanisms for increasing chlorine resistance still need to be explored [23]. Recently, molecular dynamics result revealed that the stabilizing effect of MWCNT hinders the chlorination of the PA structure [23]. Moreover, a study has prepared a uniform interconnected structure of polydopamine wrapped SWCNT film supported ultrathin PA NF membrane [118]. The CNT membrane shows long-term resistance to chlorine because of the high degree of crosslinking (Figure 6a) [118]. In addition, the CNT NF membrane displays high retention of negative charged divalent ions by size exclusion and electrostatic repulsion effect [118]. Therefore, to improve the chlorine resistance of the membrane, a large number of large-diameter CNTs can be incorporated in PA membranes, as well as the crosslinking degree between the CNTs and the polymer can be increased.

#### 2.2.4. Mechanical Performance

The mechanical performance of composited CNT membranes will also be enhanced as CNTs are introduced [158,159,160,161,162,163]. Carbon nanotubes can function well as appropriate support [158]. In graphene–nanomesh (GNM) SCWNT hybrid membranes, SCWNT webs interact strongly with GNM—a strong π−π interaction—and provide and provide mechanically strong support [159]. In addition, gradual tensile strength increases with MWCNT content [164]. Because structural integrity will reduce due to the large gap caused by the low load of CNTs [162]. In addition, the mechanical property will be strengthened with the incorporation of MWCNT in chitosan composite membranes [163]. The main reason is that the addition of MWCNTS can significantly improve the crystallinity and tensile strength of chitosan in composite membranes [163]. Moreover, the CNT interlayer can absorb [165] and store the aqueous amino solutions to promote the interfacial polymerization and provide robust mechanical support [166,167] for the NF of the thin skin layer on a microporous substrate (Figure 6b) [103]. CNTs can be used as supporting layers or interlayers to enhance the mechanical strength of the composite film.

## 3. Challenges and Outlook

### 3.1. Cost

Even though CNT researches have been conducted successfully in the laboratory over a long period, high quality and reproducible CNTs are still not available for commercial desalination applications because of their high production cost [19]. It is inspiringly expected that CNT cost will be significantly reduced in the future as more CNTs for commercial use are produced [7]. In addition, the recent development of material synthesis may enable to manufacture large quantities of high-quality and cost-efficient CNTs [19].

### 3.2. Scale-Up, Stability and Reproducibility

Conventional VACNT membranes face the challenge in controlling CNT size and creating CNT arrays of high pore density [21]. Although researchers can control the size of sub-nanometer (~0.6 nm) pores precisely in the laboratory, in terms of high permeability at close to 100% salt rejection, the actual scaling up of CNT membranes from the micrometer experimental prototype is not straightforward [21]. Scaling up CNT membranes may result in the reduction of the water permeability or salt rejection performance [21]. The development of large-area CNT membranes is expected as previous findings indicate that freestanding films can be derived from VACNTs which is easy to detach from the growth substrates and transfer multiple VACNT films onto large-size membranes or substrates [19]. However, the suitable fabrication methods for making reproducible CNT membranes are still in doubts [19]. Hence, scalable production is a feasible way to handle this challenge [19,21]. Stacking multilayered horizontal aligned CNT (HACNT) membranes can be an ideal replacement for large-scale sub-nano VACNT membranes. Moreover, interior-functionalized CNT membranes [158,166,168] are excellent choices because of their low sensitivity to variations in CNT size during physical fabrication, and their excellent desalination performance showed computationally [21]. Surface modification is another promising approach to improve the surface properties and render for functional efficiency [21].

The integration of traditional technology and CNTs and the development of advanced desalination systems also need to be considered. The main concern should be raised about the reproducibility of CNT membranes as well [19]. For practical application, the cost of replacing the module is higher than that of CNT membranes. Therefore, it is vital to evaluate the performance of CNT membranes. In addition, most of the studies have neglected long term performance, including performance drop, maintenance and cleaning [169]. Following studies should be focusing on long term performance data using real feed solutions [19].

### 3.3. Simulation and Mechanism

On one hand, some proposed CNT designs such as the rotating CNT configuration or the nonconventional shape CNT membranes require more advanced nanoscale fabrication techniques [21]. Although there is research effort ongoing, no obvious solution is found yet [21]. On the other hand, experiments at the atomic scale are still hard to be performed due to the characterizations of some nanomaterials [19]. Therefore, MD simulation has been widely applied to investigate the structure existed in the water passage across CNT membranes [19]. However, CNT membranes are known to suffer from water flux that is far below the predicted value [19]. These CNT membrane disadvantages have significantly thwarted the separation characteristics in ideal condition [19]. Consequently, most of the fascinating properties cannot be fully realized during experimental tests [19]. Moreover, it was found that variations in the surface properties can play critical roles in manipulating the measured water flux [19].

Additionally, the inconsistency highlights the challenges in fully understanding the transport mechanisms through carbon nanotubes [19]. Hence, more experiences should be designed to have a better understanding of the behaviors of carbon nanotubes [19]. It is also essential to examine the relationship between the structural properties and the synthesis conditions, transport mechanisms and eventually, the separation performance of CNT membranes [19]. From a material science point of view, understanding chemical effects will lay a solid foundation for fabricating CNTs with desired characteristics [19].

### 3.4. Compatibility and Environmental Impact

Functionalization of CNTs can improve dispersion in the matrix, and there are many ways to incorporate CNTs into a polymer matrix [7]. However, excessive modification treatment brings about degradation of the properties of CNTs and the generation of chemical waste [7]. Therefore, innovative techniques are expected to address the challenge of functionalized CNTs without or minimizing structural damage and environmental pollution [9].

It is generally recognized that raw CNTs are more toxic than functionalized CNTs because of the existence of metal catalyst [7]. Nevertheless, there are scarce studies concerning the negative impacts of CNTs on the environment as well as human [147]. Therefore, the unclear hazard of CNTs on the environment and human health must be investigated and evaluated in detail [7]. However, previous studies on the impact of CNTs are mainly based on short-term effects [147]. For this reason, future investigations with attention to the longer-term implications of CNTs can help determine the environment’s accurate toxicological profile [147]. Beyond that, necessary precautions need to be suggested in the complete procedure and implemented techniques to enable environment-friendly CNTs [7].

### 3.5. Others

To improve CNT membrane desalination performance, the pH of the solution, applied electric field and electrostatic interactions are of concern [7]. Moreover, most of the current CNT membranes are assembled on ceramic or polymer membranes, affecting the properties of CNTs [7]. Therefore, more attention should be paid to the fabrication of freestanding membranes. In addition, due to the excellent conductivity of CNTs, extensive research is needed to explore membrane distillation and capacitive deionization and the other possible applications of CNT-based membranes [7].

## 4. Conclusions

With the development of CNTs at a steady pace, CNT membranes can act as advanced material for cutting-edge desalination technology. Numerous advantages that promote CNTs in desalination, including enhanced water flux, high selectivity, antifouling and mechanical properties. Some simulations and experiments have also been performed to investigate the mechanisms and effects of CNT addition to enhance the membrane performance for water desalination.

Despite CNTs’ enormous potential for developing high-performance desalination membranes, there are still major challenges to overcome. First, CNT membranes are limited to the high cost of CNTs on a commercial scale, especially SWCNTs. In recent years, with the increase of industrial production, the unit price of commercial CNTs has been boldly dropped. In this regard, commercialization of CNT membranes for desalination technologies is becoming increasingly possible. While the integration of traditional technology and CNTs and the development of advanced desalination systems need to be considered. Second, CNT membranes still suffer from the tradeoff between removal efficiency and water permeability in spite of smooth and nonpolar interior channels of CNTs. MD simulation suggested that sub-nano (~0.6 nm) CNTs can achieve extremely rapid water penetration and ion-selective separation. However, the main problems with producing such small diameter nanotubes are not yet practical. Furthermore, large scale preparation of high-density VACNT arrays is still is challenging due to the limitations of existing preparation methods since the irregular alignment affects the membrane properties. The blockages of hydrated ions in large-diameter CNTs and the tip functionalization of CNTs are also critical issues while using VACNT membranes for desalination. Although the simulation results indicate that the CNTs with specific shapes and well-defined morphologies are expected to achieve water penetration much faster, the way of preparing such kind of CNTs is necessary to be investigated. Third, the synthesis of composited CNT membranes for desalination has attracted much attention because of its high-water permeability and antifouling properties. With the decrease of the number of graphite layers of CNTs, the agglomerate tendency of CNTs due to van der Waals forces increase. Functionalization of CNTs can address this challenge and improve dispersion in the matrix. There are also many ways to incorporate CNTs into a polymer matrix, including phase inversion, interfacial polymerization, solution mixing, spray-assisted layer-by-layer, polymer grafting, etc. Meanwhile, the improvement of salt rejection performance of functionalized CNT membranes can be attributed to the electrostatic interactions, Donnan effect and chemical interactions between CNTs and salt ions. However, excessive modification treatment results in degradation of the properties of CNTs and the generation of chemical waste. Therefore, innovative techniques are expected to achieve functionalization of CNTs without or minimizing structural damage and environmental pollution. Furthermore, the exact and complete mechanisms of water molecule transport and flow rate inside the CNTs, especially in the sub-nano scale, need to be made clear. Fourth, the potential harm of CNTs to human health and the environment needs to be studied, and necessary preventive measures must be taken even though functionalized CNTs are less toxic than the raw ones. To improve the desalination performance of CNT membranes, the pH of the solution, applied electric field and electrostatic interactions are also of concern. It is important to note that most of the current CNT membranes are assembled on ceramic or polymer membranes, which may affect the properties of CNTs. Therefore, more focus should be paid to the preparation of freestanding membranes to take full advantage of the extraordinary features of CNTs. The application of CNT in desalination filed may be broadened by undertaking the following future research:Stacking multilayered horizontal aligned CNT (HACNT) membranes are an ideal replacement for sub-nano VACNT membranes and the scale-up and high-density preparation of this HACNT membrane is also feasible.The combination of CNTs with excellent mechanical properties and nanoporous materials (e.g., single-layer nanoporous two-dimensional nanomaterials) with atomic layer thickness can achieve ultra-thin, high-permeability desalination membranes.The modification of CNTs not only improves the dispersion in the polymer matrix, but also the desalination performance and anti-pollution performance of the membranes, so it is essential to simulate and optimize the modification of CNTs.Due to the excellent conductivity of CNTs, in addition to pressure-driven CNT membranes, extensive research is needed to explore the other possible applications of CNT-based membranes such as membrane distillation and capacitive deionization.

Even though CNT membranes face some problems such as larger-scale preparation for VACNT membranes and long-term performance, further development of nanomaterials and nanotechnology will provide new solutions to deal with these issues. In general, CNTs have dramatically changed the concepts of conventional membranes concerning both preparation and performance for desalination. Not only vertically aligned CNT membranes, but also CNT composite membranes exhibit an outstanding separation performance. Overall, we expect that more advanced CNT membranes with higher water flux, better separation performance, as well as large-scale preparation, can be explored in desalination field for practical applications.

## Figures and Tables

**Figure 1 nanomaterials-10-01203-f001:**
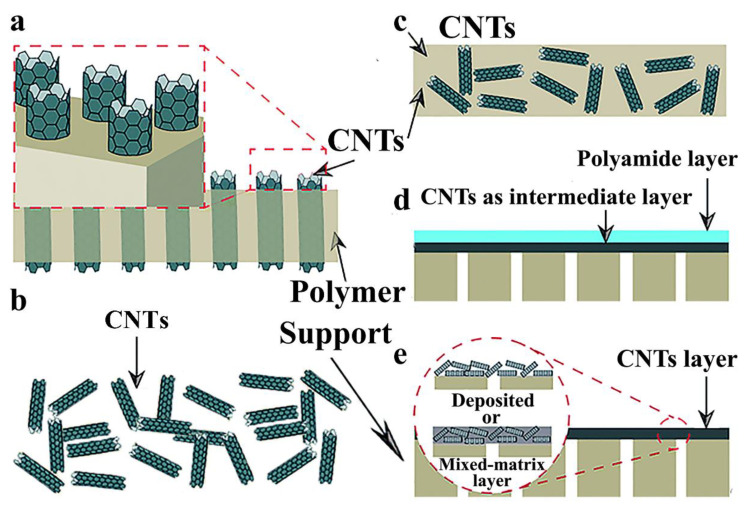
Carbon nanotubes (CNT) membranes with various structures. Freestanding CNT membranes: (**a**) vertically aligned CNT (VACNT) membranes, (**b**) buckypaper CNT membranes; composited CNT membranes: (**c**) mixed-matrix CNT membranes, (**d**) membranes with CNTs coated on the membrane surface (support) as the intermediate layer, (**e**) membranes with CNTs coated on its surface or support (Reproduced or adapted from ref. [24], with permission from The Royal Society of Chemistry, 2017).

**Figure 2 nanomaterials-10-01203-f002:**
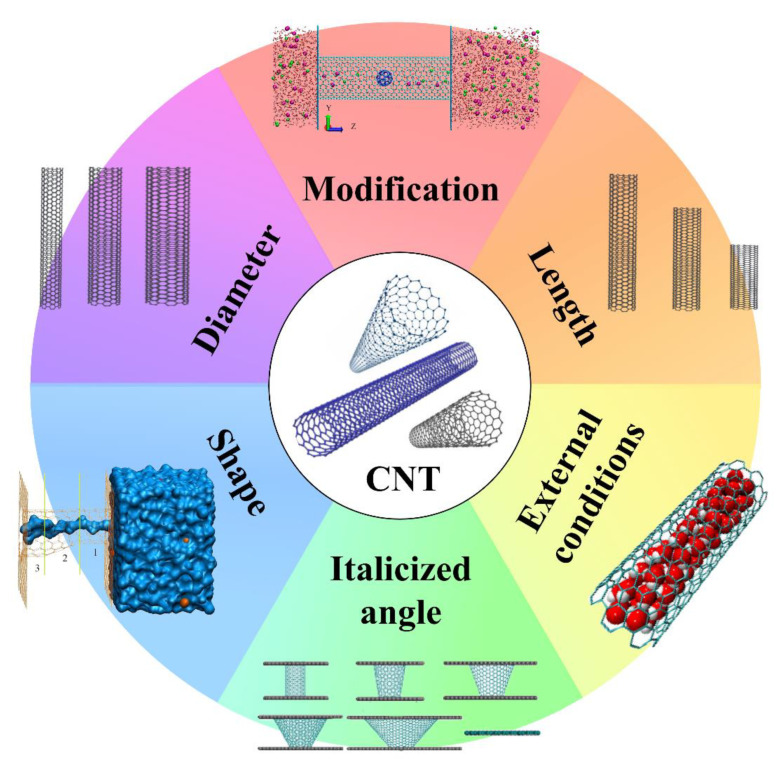
Some factors that affect the flow of water molecules in CNT channels (reproduced or adapted from ref. [35,36,37,38], with permission from Elsevier and American Chemical Society, 2017 and 2019).

**Figure 3 nanomaterials-10-01203-f003:**
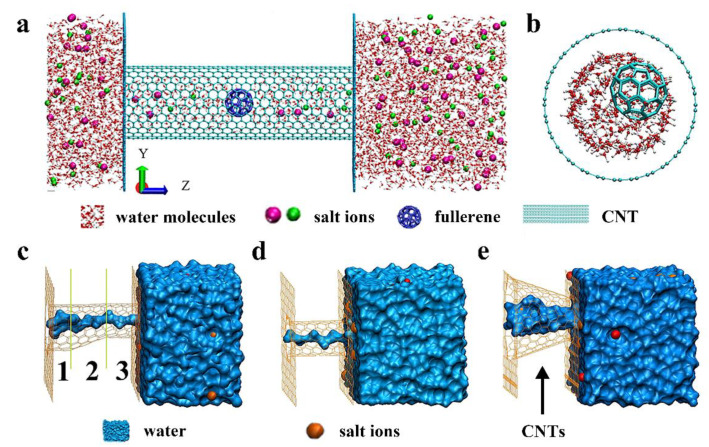
Characteristics affecting the water flow rate of CNTs. The initial structure of the F@CNT (**a**) from the side view and (**b**) the front view snapshot (Reproduced or adapted from ref. [38], with permission from Elsevier, 2019). Schematics of different shapes of CNTs; (**c**) hetero-junction CNT, (**d**) tubular (6, 6) CNT, (**e**) cone-shaped CNT (reproduced or adapted from ref. [35], with permission from Elsevier, 2017).

**Figure 4 nanomaterials-10-01203-f004:**
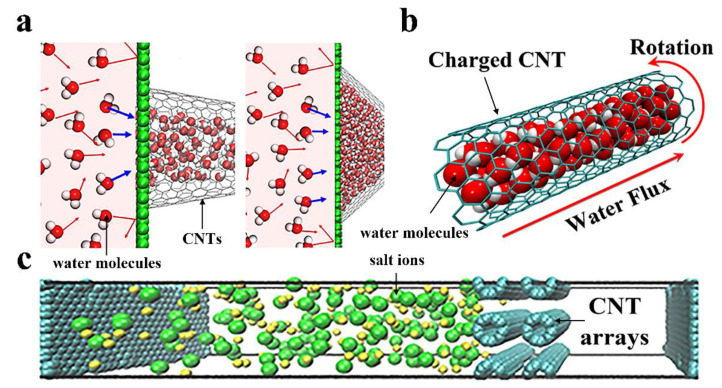
Characteristics affecting the water flow rate of CNTs. (**a**) Cartoon representation of the channel architecture effects on water permeation. CNT channels with 19.2° apex angles at left and 38.9°at right. The water molecules in the channels and at the left of the channels represent the real simulation configuration and schematic diagram, respectively (Reproduced or adapted from ref. [37], with permission from Elsevier, 2017); (**b**) diagram of the directed water transportation in rotating charged CNTs (reproduced or adapted from ref. [36], with permission from American Chemical Society, 2017); (**c**) The illustration of the CNT arrays as multilayer transverse flow membrane for desalination (Reproduced or adapted from ref. [81], with permission from Elsevier, 2019).

**Figure 5 nanomaterials-10-01203-f005:**
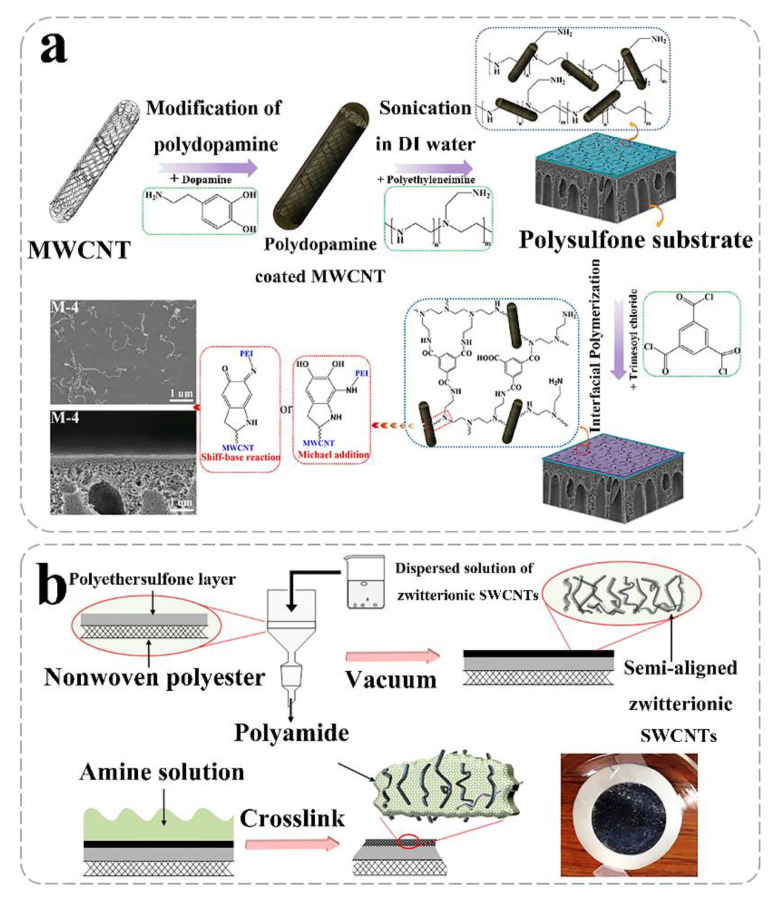
Preparation of composited CNT membranes. (**a**) Schematic preparation of positively charged nanocomposite NF membranes filled with poly(dopamine) modified MWCNTs (reproduced or adapted from ref. [105], with permission from American Chemical Society, 2016); (**b**) cross-sectional schematic of the fabrication for Z–SWNT/PA nanocomposite membrane and top view of the photograph of Z–SWNT nanocomposite membrane (Reproduced or adapted from ref. [100], with permission from Elsevier, 2016).

**Figure 6 nanomaterials-10-01203-f006:**
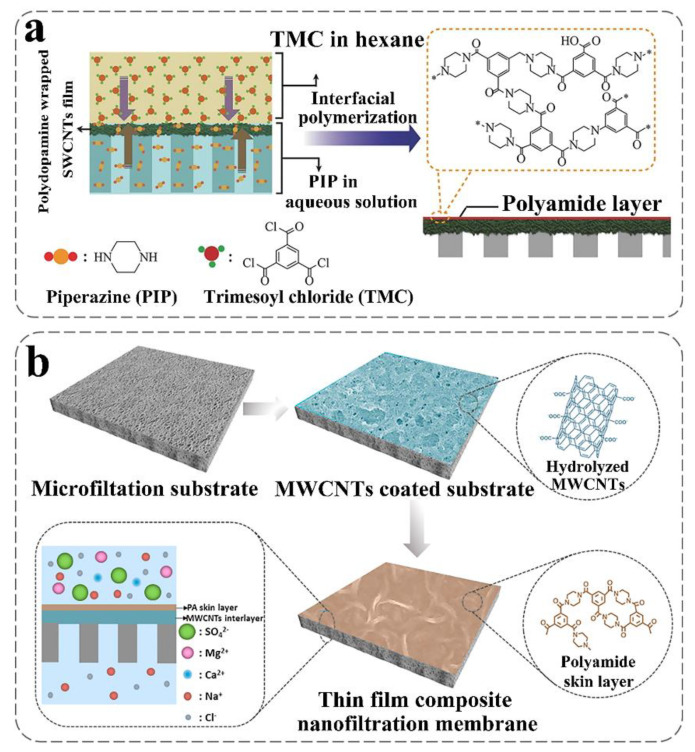
Preparation of composited CNT membranes. Schematic diagram of (**a**) interfacial polymerization process for the preparation of PD/SWCNTs film supported ultrathin PA NF membrane (Reproduced or adapted from ref. [118], with permission from Wiley–VCH Verlag, (Weinheim, Germany), 2016); (**b**) TFC membranes combining CNT intermediate layer and microfiltration support with high NF properties (reproduced or adapted from ref. [103], with permission from Elsevier, 2016).

**Table 1 nanomaterials-10-01203-t001:** Summary of the performance evaluation of differently composited CNT membranes for desalination.

Preparation Method	Composition of Membrane	Feed System	Applied Pressure	Water Flux	Membrane Rejection	The Effect of CNT Addition	Ref.
Phase inversion	MWCNT doped PVDF–HFP	2-g/L NaCl	N/A	N/A	N/A	Improved hydrophilicity	[89]
	CA + CNT	5-g/L NaCl	35 bar	N/A	94%	Increased pure water flux	[90]
	CA + CNT	58.5-g/L NaCl	N/A	18.1 LMH	N/A	Increased water flux	[91]
	CA + CNT	MgSO_4_	N/A	69.5 LMH	90.6%	Enhance permeability	[92]
	PES/CNTs + MMMs	0.2-g/L Na_2_SO_4_	4 bar	∼45.2 LMH	∼87.2%	Enhanced the water flux and salt rejection	[93]
	PSf + DDA–MWCNTs	DI water	1 bar	N/A	N/A	Improved fouling resistance	[94]
	functionalized CNT/PS	N/A	2.06 bar	600 LMH	N/A	Enhanced pure water permeation	[95]
Interfacial polymerization	PS-20 + MPD + TMC + MWCNT	2-g/L NaCl	225 psi	43 LMH	99%	Enhanced water flux and pure water permeance	[96]
	MWCNT–TNT hybrid + TMC + MPD + PS	2-g/L NaCl	15 bar	0.74 LMH/bar	97.97%	Experienced high-water permeability	[97]
	MWCNTs + PA	2-g/L NaCl	15 bar	28.9 LMH	96.7%–97.8%	Improved water flux and better antifouling performance	[98]
	PA + CNT + PSf	2-g/L NaCl	15.5 bar	46.2 LMH	97%	Enhanced water flux	[99]
	ZfCNTs + PA	2-g/L Na^+^	2.41 MPa	34.7 LMH	Above 98%	Improved surface biofouling resistance and salt rejection	[100]
	TA–MWNTs + PA	0.71-g/L Na_2_SO_4_	0.6 MPa	31.4 LMH	N/A	Optimized water flux and salt rejection	[101]
	SMWCNT	1-g/L Na_2_SO_4_	0.6 MPa	13.2 LMH/bar	96.8%	Enhanced water flux and antifouling ability	[102]
	PES + CNT	1-g/L Na_2_SO_4_	0.6 MPa	105.4 LMH	~95%	Provided a robust mechanical support	[103]
	PDA/CNTs + TMC + PSf	0.3-g/L alginate	N/A	12.4 LMH	N/A	Antifouling capacity	[104]
	Modified PDA-MWCNTs + PEI + TMC	0.5-g/L ZnCl_2_	6 bar	15.32 LMH/bar	93%	Improved water permeability	[105]
	PES + CNT + PA	Na_2_SO_4_ and MgSO_4_ solution	5 bar	21 LMH/bar	98.3%	Exhibited an excellent water flux and comparable salt rejection	[106]
	PA + MWCNT	3.5 wt% saline water	5 MPa	1.7 (m^3^/m^2^·day^−1^)	90 wt%	Antifouling nature, chlorine resistance	[107]
	PA + MWCNT	N/A	N/A	N/A	N/A	Improved chlorine resistance, antifouling and desalination performance	[108]
	PES/SPSf/O–MWCNT	brackish waters	3 bar	30.2 LMH/bar	25 (NaCl/Na_2_SO_4_ selectivity)	Enhanced water permeability and salt rejection	[109]
Interfacial polymerization + phase inversion	fCNT + PES	35.1-g/L NaCl	N/A	11.98 LMH	N/A	Contributed to the water flux and diminish of alginate fouling	[110]
	TMC + MPD + fCNT	1-g/L NaCl	10 bar	20.23 LMH	72.3%	Enhanced water permeability	[22]
Electrospray-assisted interfacial polymerization	CNTs + PA	1-g/L NaCl	4 bar	96.8 LMH	89.6%	Improved in water flux	[111]
Dissolution casting method	CA/PEG400 + SE–MWCNT	1-g/L NaCl	4 bar	0.61 LMH	99.8%	Improved the salt rejection performance	[112]
Surface Grafting and nanoparticle incorporation	Hydrophilic monomer + c-MWCNT	2-g/L NaCl	from 20 bar to 15 bar	N/A	Above 96%	Reduced fouling	[113]
ATRP + Interfacial polymerization	fMWCNTs + PAAm + PA	2-g/L NaCl	1.55 MPa	66.9 LMH	95.3%	Improved membrane separation and antifouling capability	[114]
Electrophoretic deposition + chemical reduction	RGO + CNTs	5.85-g/L NaCl	1 bar	40.4 ± 3.7 LMH	94.0% ± 1.9%	Increase water flux	[115]
Filtration-assisted interfacial polymerization	PVDF + CNT + PA	117-g/L NaCl	0.3 MPa	0.54 ± 0.08 LMH	(91.3 ± 0.4)%	Improved the water flux	[116]
Brush painting	SWCNT + PA + PES	1-g/L Na_2_SO_4_	6 bar	40 LMH	96.5%	Improved permeability	[117]
Polymerization	PDA/SWCNTs	1-g/L Na_2_SO_4_	6 bar	32 LMH/bar	95.9%	Long-term chlorine tolerance property	[118]
Chemically synthesized	PPy-raw or PPy-oxidized MWCNTs	2-g/L Na_2_SO_4_	1 MPa	88.9 LMH	96.6 wt%	Improved in the flux	[119]
ATRP	ZCNTs + poly(4-styrenesulfonic acid) + P4VP	Model brackish water	0.6 MPa	14.9 ± 0.5 LMH	5.6 ± 0.8 (MgSO_4_/NaCl selectivity)	Enhanced water permeability	[120]

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
