# Peer review of "Recent Advances in Applications of Carbon Nanotubes for Desalination: A Review"

_nanomaterials, 2020, doi:10.3390/nano10061203_

Round 1

Reviewer 1 Report

This paper focused on the literature review in the area of carbon nanotubes application for water desalination. The article is well written and easily understandable. However, there is a lack of critical comments and discussion regarding the presented results in the cited works. The Authors should indicate what the new element is in the present study. Therefore, the last paragraph of the introduction should be corrected, and the element of novelties should be highlighted. The impact of some crucial parameters affecting the physicochemical properties of CNTs e.g., CNT length, the diameter should be discussed in more detail.

Reviewer 2 Report

This paper reviewed advantages, current obstacles and breakthrough points of carbon nanotubes (CNTs) as applying in membrane water treatment processes. This paper summarized categories of CNT membranes, important properties of CNTs and how these properties affected to membrane performance enhancement, various CNT membrane synthesis methods, CNT membrane applications, and current major problems for CNT membrane industrialization. Also, this paper listed over 30 studies regarding CNT membrane for desalination, showing its permeability and salt rejection under various operating conditions.

Nevertheless, this paper does not seem helpful for readers who are interested in CNT membranes. Authors reviewed over 150 papers, but lack of explanation was shown for how CNT membrane enhanced membrane performance in each study. Also, the main advantage of CNT membrane, fouling resistance, needs more explanation related to antimicrobial/toxicity of CNTs itself and how CNTs in membrane mitigate fouling effect.

Although this paper well-summarized effect of CNTs in membrane and its advantages, major revision seems necessary for potential readers of this paper who desire to understand the CNT membranes and its advantages/challenges.

Specific Comment

  1. Although introduction well summarized overall contents of this paper, it seems too brief. Usually, review papers’ introduction contains brief history of development, advantages, obstacles, etc. In this paper, introduction is too focused on the advantages of CNT membranes, while no explanation of history of development and what kind of desalination processes used CNT membranes. Please introduce more details of this technology.
  2. In section 1. Mechanisms and influence factors for desalination, various CNT characteristics and its affect to CNT membranes are explained. However, CNTs characteristics could differently affect to the type of CNT membranes, as authors kindly mentioned in section 2. In each influence factor section, please mention the type of CNT membranes and how the influence factors affect to the membrane.
  3. CNT’s hydrophobic characteristic is one of key influential factors in CNT membranes, but no specific explanation of hydrophobic characteristic was shown in the paper. This could be explained under The modification of CNTs section in 2.1.1/ the characteristics affecting the water flow rate of CNTs
  4. Table 1. Summery of the performance evaluation of different composited CNT membranes for desalination showed various study results of CNT membranes. Although many studies are summarized, it was hard to recognize the main effect of each study with different pressure/concentration/flux units of feed system and applied pressure.
  5. Also, in Table 1, it was uncertain that all summarized studies were for RO desalination. If studies were applied for different water treatment processes such as MF/UF or FO, comparing water flux and salt rejection of each study seemed not necessary. Please sub-categorize each study’s result by process type, primary effect, or membrane synthesis type.
  6. Explanation of each study’s performance enhancement by CNT was barely shown. The main contents of this paper should be technological development of CNT-induced membranes, explaining details of each study seemed important. However, this paper’s explanation was more focused on general properties of CNTs and its affect to the membrane.
  7. Many studies are already published regarding CNT membranes’ fouling effect. Anti-fouling property is one of key factors of CNT membranes; however, this paper only briefly explained fouling resistance of CNT membranes. Please add more details of fouling resistance part such as possible anti-fouling mechanisms of CNT membranes.
  8. Overall, clear sub-categorization and each category’s key enhancement mechanism should be explained. Also, CNT membranes’ applied desalination should be mentioned to definite compare of each study.

Round 2

Reviewer 1 Report

The manuscript has been improved.

Reviewer 2 Report

The revised version with comments have now been improved. It still requires English proofreading from native or other proofreading system and would preferably write more concisely. 
